# An Extended Reservoir of Class-D Beta-Lactamases in Non-Clinical Bacterial Strains

Valérian Lupo,[a,b] Paola Sandra Mercuri,[b] Jean-Marie Frère,[b] Bernard Joris,[b] Moreno Galleni,[b] (ID) Denis Baurain,[a] Frédéric Kerff[b]

[a]InBioS-PhytoSYSTEMS, Eukaryotic Phylogenomics, University of Liège, Liège, Belgium
[b]InBioS, Center for Protein Engineering, University of Liège, Liège, Belgium

**ABSTRACT** Bacterial genes coding for antibiotic resistance represent a major issue in the fight against bacterial pathogens. Among those, genes encoding beta-lactamases target penicillin and related compounds such as carbapenems, which are critical for human health. Beta-lactamases are classified into classes A, B, C, and D, based on their amino acid sequence. Class D enzymes are also known as OXA beta-lactamases, due to the ability of the first enzymes described in this class to hydrolyze oxacillin. While hundreds of class D beta-lactamases with different activity profiles have been isolated from clinical strains, their nomenclature remains very uninformative. In this work, we have carried out a comprehensive survey of a reference database of 80,490 genomes and identified 24,916 OXA-domain containing proteins. These were deduplicated and their representative sequences clustered into 45 non-singleton groups derived from a phylogenetic tree of 1,413 OXA-domain sequences, including five clusters that include the C-terminal domain of the BlaR membrane receptors. Interestingly, 801 known class D beta-lactamases fell into only 18 clusters. To probe the unknown diversity of the class, we selected 10 protein sequences in 10 uncharacterized clusters and studied the activity profile of the corresponding enzymes. A beta-lactamase activity could be detected for seven of them. Three enzymes (OXA-1089, OXA-1090 and OXA-1091) were active against oxacillin and two against imipenem. These results indicate that, as already reported, environmental bacteria constitute a large reservoir of resistance genes that can be transferred to clinical strains, whether through plasmid exchange or hitchhiking with the help of transposase genes.

**IMPORTANCE** The transmission of genes coding for resistance factors from environmental to nosocomial strains is a major component in the development of bacterial resistance toward antibiotics. Our survey of class D beta-lactamase genes in genomic databases highlighted the high sequence diversity of the enzymes that are able to recognize and/or hydrolyze beta-lactam antibiotics. Among those, we could also identify new beta-lactamases that are able to hydrolyze carbapenems, one of the last resort antibiotic families used in human antimicrobial chemotherapy. Therefore, it can be expected that the use of this antibiotic family will fuel the emergence of new beta-lactamases into clinically relevant strains.

**KEYWORDS** antimicrobial resistance, beta-lactamase, phylogenetic classification, sequence clustering, carbapenemase, OXA

Beta-lactamases are the main enzymes responsible for the resistance of bacteria to beta-lactams, which are among the most common antibiotics used in the fight against pathogenic bacteria. Even before the structure of penicillin was known, Abraham and Chain (1) described "An enzyme from bacteria able to destroy penicillin" and, in the late 1940s and early 1950s, the staphylococcal beta-lactamase became an important source of clinical problems solved by the introduction of methicillin (2, 3). Later, an ever increasing number of these hydrolases have been identified. These can be classified into four classes based on

**Ad Hoc Peer Reviewer** (ID) Peter Oelschlaeger, Western University of Health Sciences

Address correspondence to Denis Baurain, denis.baurain@uliege.be, or Frédéric Kerff, fkerff@uliege.be.

The authors declare no conflict of interest.

their primary structures. Classes A, C, and D are active-serine enzymes (4) while class B consists of metallo-proteins whose active site usually contains 1 or 2 $Zn^{++}$ ions (5, 6).

Beta-lactamases of classes A and D exhibit a very high diversity of amino acid (AA) sequences, with only a very little number of conserved residues within each class (e.g., 29 residues are conserved within class-A beta-lactamases) (7). It is nearly impossible to establish clear relationships between AA sequences and the ability to hydrolyze specific classes of beta-lactam antibiotics. Indeed, it is well known that a single mutation can alter this activity profile in a significant manner (8, 9). Moreover, the literature contains numerous disagreements and errors concerning the kinetic parameters of various enzymes (10). This is probably in part because these parameters are often determined under different experimental conditions and the studied enzymes are not always pure. In consequence, even though clinicians are more interested in specificity profiles, the AA sequences remain the primary tool for proposing a classification of beta-lactamases, as in the case of the Beta-Lactamase Database (BLDB; http://www.bldb.eu/) (11). Concerning class D beta-lactamases, the situation is complicated by the fact that these enzymes can dimerize, which sometimes modifies the activity (12) and that carboxylation of the first conserved motif lysine also increases the activity in most cases (13). Inversely, loss of $CO_2$ during turn-over of the substrate results in "substrate-induced inactivation," a phenomenon already observed by Ledent et al. (14).

The first two identified class D beta-lactamases exhibited a number of features that differed from those of nearly all beta-lactamases known at the time, notably the ability to efficiently hydrolyze oxacillin and other isoxazolyl penicillins. For this reason, they were named OXA-1 and OXA-2. Unfortunately, it was then decided to name the further class D beta-lactamases homologs "OXA" plus sequence (i.e., increasing) number that follows the chronological order of identification (15). This was sometimes done in spite of a sequence identity below 30% and/or (10) more similarity with the BlaR receptor than with other class D beta-lactamases (16). Class D beta-lactamases were first identified as plasmid-encoded proteins but the corresponding genes were later found to reside on the bacterial chromosome too (10).

Similarity searches using the OXA-2 AA sequence as a query revealed homologous primary structures of unknown function or without true beta-lactamase activity, such as YbxI/BSD-1 in *Bacillus subtilis* (17, 18), or even devoid of any beta-lactamase activity, such as the C-terminal domain (CTD) of the BlaR penicillin-receptor involved in the induction of a class A beta-lactamase in *Bacillus licheniformis* and *Staphylococcus aureus* (19). In the present study, proteins containing a class-D beta-lactamase domain will be further referred to as the "OXA-domain family." Among those, "DBL" will be reserved to demonstrably active class-D beta-lactamases, while characterized class-D beta-lactamase homologs of low activity or with a different function will be termed "pseudo-DBL" proteins. Finally, "DBL-homolog" proteins will define the union of DBL, pseudo-DBL proteins, and other homologs not yet characterized.

It is clear that our present knowledge of the OXA-domain family is biased toward clinically relevant DBLs. The analysis of whole genome sequences of isolated bacteria and metagenome-assembled genomes highlighted that non-pathogenic and environmental bacteria can also harbor beta-lactamase-encoding genes, and thus may behave as reservoirs of emerging new resistance genes identified in nosocomial strains (20, 21). It is likely that these bacteria, which, in many cases, were never exposed to synthetic or semi-synthetic beta-lactams used in human health care or animal husbandry, can encounter other beta-lactam-producing microorganisms in their natural environment and, over the ages, have acquired beta-lactamase genes in their "struggle for life" (22). A significant example could be the carbapenems that can be produced by some *Streptomyces* species (23), probably resulting in the appearance of the carbapenemases that were later transferred to clinical strains (24, 25). The large heterogeneity of the resistance gene repertoire present in bacteria challenges the efficiency of antimicrobial chemotherapy. It also underlines the need to develop new analytical methods

allowing a clear and rapid identification of potential new resistance pathways including enzymes that can inactivate both old and new antibiotics.

The goals of the present article were to explore genomic databases to discover how widespread the class D beta-lactamase gene and its homologs were throughout the microbial world and to propose a sequence-based classification of the members of the OXA-domain family derived from their phylogenetic relationships. Starting from 80,490 genomes, we identified a total of 24,916 DBLs and DBL-homolog sequences, which we classified into 64 clusters of proteins. Furthermore, we synthesized and expressed 10 gene sequences sampled from 10 clusters devoid of characterized members and conducted a survey of their activity. This revealed that three of them had an oxacillinase activity, including two able to hydrolyze imipenem, reminding us how environmental bacteria represent an enormous reservoir of resistance factors that can be transferred to clinical strains.

## RESULTS

**Enlarging the OXA-domain family taxonomic distribution.** According to the BLDB (as of July 2019), the 810 described DBL-homologs (including both DBL and pseudo-DBL proteins) have been isolated from bacteria belonging to five different phyla: Proteobacteria (583 Sequences), Spirochaetes (14), Firmicutes (9), Bacteroidetes (1), Fusobacteria (1) and also from some marine metagenomes (2). Two-hundred sequences have no source organism and are all plasmid-encoded. Most of these DBL-homologs are found in Proteobacteria, essentially in the genera *Acinetobacter* (411 sequences) and *Campylobacter* (91), which are part of the Gammaproteobacteria and Epsilonproteobacteria, respectively. Some are also found in Betaproteobacteria (41) but not in the other Proteobacteria classes.

A HMM profile constructed from an alignment of 470 DBL from NCBI Pathogen Detection server allowed us to identify 24,916 OXA-domain family AA sequences distributed across 20,342 organisms (on a total of 80,490 screened genome assemblies found in NCBI RefSeq). Nearly all those organisms (99.4%) belonged to the aforementioned five phyla, whereas the small remaining fraction (0.6%) came from eight additional bacterial phyla: Cyanobacteria (65 Sequences), Actinobacteria (36), Chlorobi (10), Chlamydiae (6), Verrucomicrobia (6), Chloroflexi (2), Balneolaeota (1), and planctomycetes (1). Moreover, some sequences were identified in additional classes of proteobacteria: Alphaproteobacteria (Holosporales) and *Deltaproteobacteria* (Desulfovibrionales). In contrast, no sequences of the OXA-domain family were found in Archaea. In this work, we wanted to characterize the protein sequences themselves and, to do so, we deduplicated the 24,916 sequences and observed that they represented only 3,510 unique sequences (i.e., 100% identical at the AA level), indicating that many of them were multispecies enzymes. Indeed, it is known that the NCBI RefSeq database is unevenly biased toward clinical strains (26). Hence, 3459 of the unique sequences (98.5%) were found in several species of the same genus (e.g., WP_001046004.1 was found in 952 *Acinetobacter* species) while 51 unique sequences (1.5%) were found in more than one genus. These results show that the redundancy is mostly due to the number of species in NCBI RefSeq belonging to the same genus. In a second step, these 3,510 unique sequences were deduplicated at a global identity level of 95%, and the 1,413 resulting sequences (hereafter termed "representative" sequences) were used to infer a phylogenetic tree (see Materials and Methods).

**OXA-domain family proteins include BlaR homologs.** A distribution of sequence length showed that the 24,916 OXA-domain family sequences formed three populations, one shorter than 350 AAs with an average size of 271 AAs (typical DBL length), one longer than 550 AAs with an average size of 587 AAs (typical BlaR membrane receptor length) and one intermediate-length population with an average size of 449 AAs (Fig. 1a). Mapping sequence length onto the tree revealed that representative sequences of intermediate length are scarce (five sequences) and not clustered, whereas long sequences do cluster in two distinct groups (Fig. 1b). A sequence similarity analysis showed that three of the five intermediate-length sequences are actually

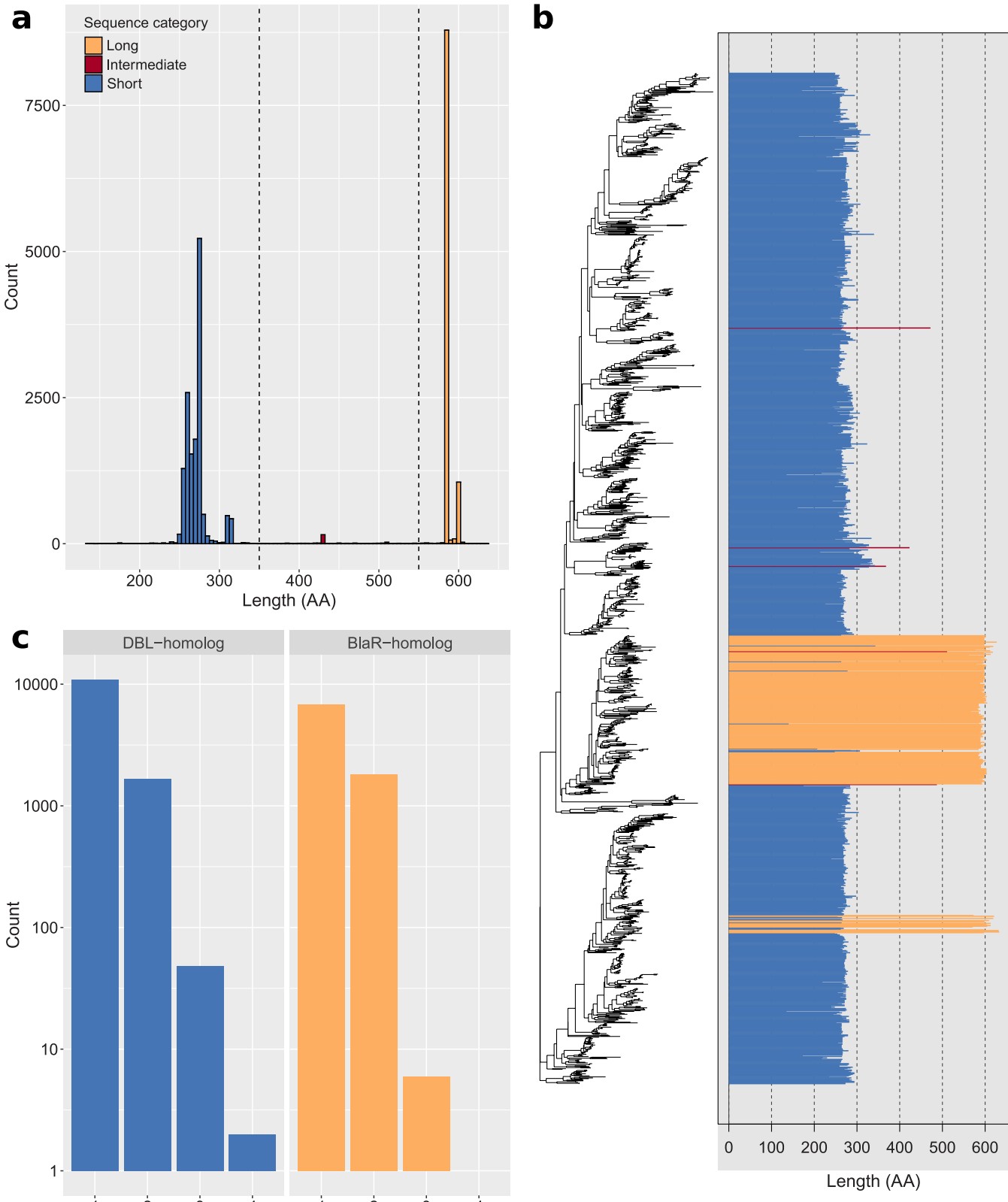

**FIG 1** Classification of OXA-domain family protein sequences as DBL-homologs or BlaR-homologs. (a) Length distribution of the 24,916 OXA-domain family protein sequences. Sequences shorter than 350 AAs are colored in blue, sequences longer than 550 AAs are in orange, while sequences between 350 and 550 AAs are in red. (b) Length distribution of the representative sequences mapped onto the phylogenetic tree. The tree was constructed from a matrix of 1,413 representative sequences × 188 unambiguously aligned AAs using RAxML under the LG+F+G4 model. (c) Distribution of the number of DBL-homolog and BlaR-homolog sequences per organism. Blue bar plots represent DBL-homolog sequences while orange bar plots represent BlaR-homolog sequences. The $y$ axis is in $\log_{10}$ units.

DBL-homologs while two are BlaR homologs. Regarding the two groups of long sequences, the larger one is formed of sequences found in Firmicutes, with a majority in *Staphylococcus*, *Clostridioides*, and *Bacillus*. According to the annotation results, these sequences are actually BlaR homologs. The second group contains sequences found in Oxalobacteraceae (Betaproteobacteria) and annotation results at first showed no close similarity with DBL nor BlaR. However, detailed *in silico* functional analysis (InterProScan and pepwindowall; Data set S1) eventually revealed that 14 of these representative sequences indeed have a class D active site and the BlaR1 peptidase M56 domain, whereas three have both a class D active site and a class C beta-lactamase active site, like in the LRA-13 fusion enzyme (20), but these exhibit a low sequence identity to the latter (around 60%). To facilitate subsequent discussion, the three intermediate-length DBL homologues and the three OXA-class C fusion proteins were considered as DBL-homologs, whereas the two intermediate-length sequences more similar to BlaR and the two groups of long sequences were considered as BlaR-homologs.

In Firmicutes, beside the 10,496 BlaR-homologs, we also found 1,383 DBL-homologs. According to the annotation results, 374 are homologous to low-activity pseudo-DBL proteins found in *Bacillus* (17, 18) and 956 are homologous to the two intrinsic pseudo-DBL (CDD) of *Clostridium difficile* (27).

In general, surveyed bacteria possess only either one DBL-homolog protein (9,964 strains) or one BlaR-homolog protein (5,874 strains). In 1,665 and 1,813 strains, we found two DBL-homologs or two BlaR-homologs, respectively, and rarely more than two DBL-homologs (27) or BlaR-homologs (5). In addition, 963 strains simultaneously possess one DBL-homolog and one BlaR-homolog, while 10 strains show more than one DBL-homolog and one BlaR-homolog or the opposite (Fig. 1c). Interestingly, strains that harbor more than one DBL-homolog (ignoring BlaR homologs) mostly belong to Pseudomonadales, and more specifically the genera *Acinetobacter* and *Pseudomonas*.

**Gene genetic context.** Among the 24,916 DBL-homolog and BlaR-homolog protein sequences initially identified, only 23,833 corresponding genes (found on 23,093 contigs) could actually be fetched from complete genomes. Three reasons explain the 1,083 missing sequences: (i) the genome has been suppressed during the study, (ii) the sequence has been suppressed or removed at the submitter's request and could not be found in the genome annotation (gff) file, and (iii) no link between the protein and any gene exists in the NCBI. According to the NCBI annotation pipeline, the contigs are classified "chromosome" in 960 cases, "plasmid" in 273 cases and "genomic" in 21,860 cases. This rather uninformative "genomic" classification led us to predict the genetic context of each OXA-domain family protein sequence using the dedicated PlasFlow pipeline. With this strategy, 15,515 contigs were classified as "chromosome" (67.2%), 5,660 as "plasmid" (24.5%), whereas 1,918 remained unclassified. These unclassified contigs correspond to 9.20% of the DBL-homolog genes (1,327 cases) and 5.79% of the BlaR-homolog genes (608 cases). In addition, 1,177 contigs were congruently classified (either as chromosome or plasmid) by both pipelines, and only four had different labels, thereby confirming the accuracy of PlasFlow for contig classification. DBL-homolog and BlaR-homolog genes are mostly chromosome-encoded (Fig. 2a), with 10,078 (69.89% of DBL-homolog genes) and 6,153 (58.62% of BlaR-homolog genes) cases, respectively, whereas the genes are plasmid-encoded in 2,159 and 3,508 cases, respectively. Thus, 14.97% of DBL-homolog genes and 33.42% of BlaR-homolog genes lie on a plasmid.

The majority of bacteria carrying a DBL-homolog gene on a plasmid belong to six genera of Gammaproteobacteria: *Acinetobacter* (724), *Klebsiella* (590), *Escherichia* (212), *Shigella* (200), *Enterobacter* (196), and *Pseudomonas* (85). The remaining plasmid-encoded sequences are distributed across the other classes of Proteobacteria (Alpha- [35], Beta- [46], and Gammaproteobacteria [61]), while a few can also be found in Firmicutes (6), Cyanobacteria (3), and Bacteroidetes (1).

To assess the transfer potential of DBL-homolog and BlaR-homolog genes, and therefore the propensity of emergence of a new resistance, we looked for transposase genes in the vicinity of these genes (Fig. 2b). We noticed that DBL-homolog and BlaR-homolog

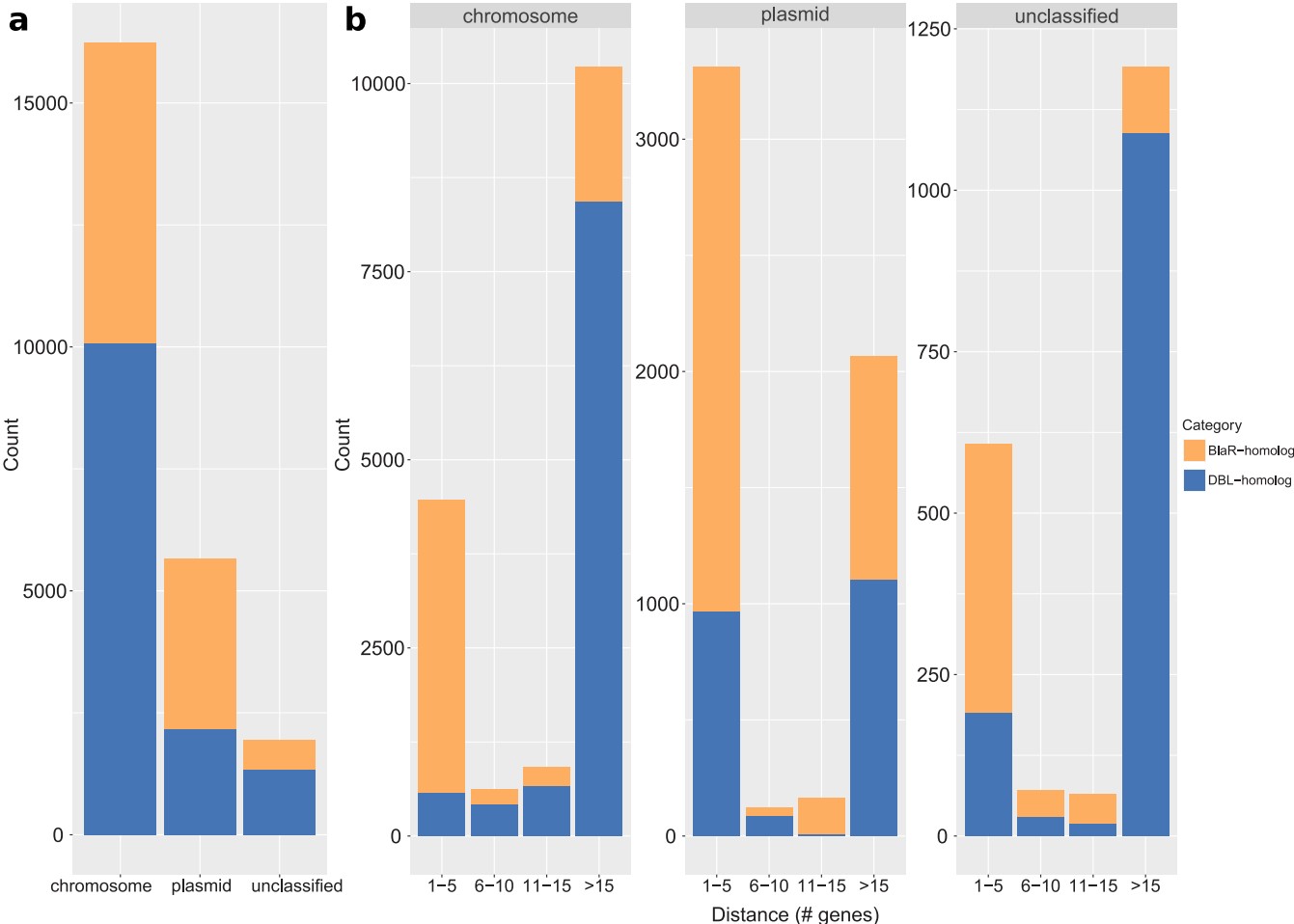

**FIG 2** DBL-homolog and BlaR-homolog genes in their genetic context. (a) Distribution of DBL- and BlaR-homolog genes according to the type of encoding molecule. (b) Distribution of the distances between DBL- and BlaR-homolog genes and transposase genes across the classified contigs. The distance is measured as a range of genes centered on the gene (DBL- or BlaR-homolog) of interest. DBL- and BlaR-homolog genes are colored in blue and orange, respectively.

genes are either close to transposase genes (distance from one to five genes) or very distant (more than 15 genes) in each genetic context. Concerning BlaR-homologs, 63.4% of the genes are close to at least one transposase gene when chromosome-encoded and 67% when plasmid-encoded. Regarding DBL-homolog genes, only 5.6% and 44.7% are close to transposase genes on chromosomes and plasmids, respectively. The majority of DBL-homolog genes encoded on chromosomes near a transposase gene (568) are found in *Acinetobacter* (395), which is also the genus in which we identified most DBL-homologs (see section OXA-domain family proteins include BlaR homologs). However, when the genes are plasmid-encoded, those close to a transposase gene (965) are mainly found in *Klebsiella* (345), then *Acinetobacter* (189), *Shigella* (177), and *Escherichia* (112). Furthermore, three DBL-homolog genes in Cyanobacteria (two on a chromosome and one on plasmid) and one chromosome-encoded gene in Balnealaoeta are close to a transposase gene, which suggests that they might have been acquired by gene transfer. In contigs not classified by PlasFlow, we observed a higher prevalence of DBL-homolog genes than BlaR-homolog genes, and these DBL-homologs are very distant from transposase genes. As this pattern is similar to the pattern observed for chromosomes (Fig. 2b), it indicates that unclassified contigs likely correspond to chromosomes.

**Signal peptide and transmembrane segment prediction.** Most DBL-homolog sequences are characterized by a signal peptide (SP), as predicted by SignalP (Table 1). The Sec, Lipo, and Tat SPs were identified in 65%, 22%, and 3% of DBL-homolog unique sequences, respectively (see "Enlarging the OXA-domain family taxonomic distribution").

**TABLE 1** Distribution of predicted signal peptides in DBL-homolog and BlaR-homolog unique sequences further broken down by the number of predicted transmembrane (TM) domains (0, cytoplasmic, 1, monotopic, > 1, polytopic)

| | DBL-homologs | | | | BlaR-homologs | | | |
|---|---|---|---|---|---|---|---|---|
| | Signal peptide (SP) | | | No SP | Signal peptide (SP) | | | No SP |
| # TM | Sec | Lipo | Tat | Other | Sec | Lipo | Tat | Other |
| 0 | 1,660 | 587 | 83 | 195 | 0 | 0 | 0 | 0 |
| 1 | 70 | 4 | 2 | 49 | 0 | 0 | 0 | 1 |
| > 1 | 0 | 0 | 0 | 11 | 2 | 1 | 0 | 845 |

The rest of the sequences (OTHER-SP, 9%) are either transmembrane proteins or have no SP. DBL-homolog sequences with a Sec-SP are mainly found in the genera *Pseudomonas*, *Burkholderia*, *Campylobacter*, *Klebsiella*, and *Legionella*, while DBL-homolog sequences with a Lipo-SP were mostly identified in *Acinetobacter* and *Leptospira*. DBL-homologs with a Tat-SP seem to be more specific to Alphaproteobacteria (*Bradyrhizobium*) whereas the "OTHER-SP" prediction is mainly associated with intrinsic pseudo-DBLs (CDD-1 and CDD-2 enzymes) of *Clostridioides* (27).

Beside signal peptide prediction of SignalP, the transmembrane segment (TM) prediction was used to distinguish between membrane proteins and cytoplasmic proteins. Whenever a SP is predicted in DBL-homolog sequences, the TM prediction indicates no TM or, rarely, one TM domain (monotopic) (Table 1). When no TM domain is detected, it may indicate that the corresponding DBL-homolog is excreted outside the cell or into the periplasmic space (for diderm bacteria). In contrast, when one TM domain is predicted, the protein is more likely to be anchored in the cytoplasmic membrane. In the majority of DBL-homologs with no SP predicted, no TM domain is detected, and these are possibly cytoplasmic proteins. Nevertheless, some exceptions exist, with 60 unique sequences (on 255 unique DBL-homolog proteins with no SP) presenting one or more TM domains (polytopic), a configuration which remains to be explained.

Almost all the BlaR-homolog proteins have no SP predicted and have, as expected, more than one TM domain (Table 1). However, only three polytopic proteins were predicted with Sec-SP or Lipo-SP instead of OTHER-SP. This can be explained by a wrong attribution by SignalP. Indeed, SignalP gives a probability for each possible SP and then chooses the highest value for the prediction but, for those sequences, the probabilities for OTHER-SP and Sec-SP/Lipo-SP are both close to 0.5.

**Prevalence of DBL-homolog genes in clinical strains.** Acquired resistance in clinical bacterial strains is a very important concern, but determining the clinical origin of a given bacterial isolate only based on the metadata of the corresponding genome assembly is still challenging to automate at a large scale. Indeed, BioSample reports from the NCBI can contain such information but these remain difficult to analyze due to the lack of a controlled vocabulary. To overcome this difficulty, we used a script that standardizes all the words of a BioSample report. Thus, 20,317 BioSample accessions were associated with the 20,342 bacterial assemblies containing DBL-homolog or BlaR-homolog genes for a total of 223 unique standardized words. Note that 4,658 BioSample reports did not contain any word. BioSamples with a positive clinical score (see Materials and Methods for details) were considered as clinical strains while those with negative scores were not. Furthermore, we decided to not classify BioSamples with a null score (essentially due to the aforementioned lack of words). Using this strategy, 3,192 bacteria were classified as *clinical*, 2,810 as *non-clinical*, and 14,340 could not be classified. Around 28% of gene sequences belong to classified strains, of which 55% clinical strains and, among those "clinical genes," 73% are DBL-homolog sequences (Table 2). Clinical DBL-homolog genes encoded on a plasmid are exclusively present in Gammaproteobacteria, mostly in *Acinetobacter* (176), *Klebsiella* (135), and *Enterobacter* (93), while DBL-homolog genes encoded on chromosomes are mostly found in Proteobacteria

**TABLE 2** Distribution of DBL- and BlaR-homolog sequences in clinical, non-clinical and unclassified strains, further broken by type of encoding molecule (chromosome, plasmid, or unclassified)

| Encoding molecule | Clinical | | Non-clinical | | Unclassified | |
|---|---|---|---|---|---|---|
| | DBL-homologs | BlaR-homologs | DBL-homologs | BlaR-homologs | DBL-homologs | BlaR-homologs |
| Chromosome | 2,080 | 459 | 1,716 | 696 | 6,282 | 4,998 |
| Plasmid | 512 | 515 | 150 | 305 | 1497 | 2,688 |
| Unclassified | 234 | 67 | 222 | 36 | 871 | 505 |

(2,016) and some in Firmicutes (43), Bacteroidetes (10), Spirochaetes (8), Actinobacteria (2), and Verrucomicrobia (1).

**Clustering and DBL-homolog selection.** Over 600 combinations of clustering parameters were tested on the OXA-domain family phylogenetic tree (see SQL database) and the clustering with the highest entropy and the lowest number of singletons (i.e., clusters of size one) was retained ($x$ set at 0.20 and inflation at 1.5; see Materials and Methods). This specific clustering solution has a computed entropy of 0.762 and a score of 0.52. It contains 64 clusters, including 19 singletons, with the larger cluster having 207 representative putative sequences (cluster 15) among 1,413 (Table S1). In general, there is little taxonomic diversity within each cluster. Indeed, the majority of these clusters (28) contain sequences from organisms belonging to the same phylum or class.

Annotating the unique sequences using BDLB reference sequences at an identity threshold set to 100% (see Materials and Methods) allowed us to tag 340 unique sequences, corresponding to 307 reference sequences (304 DBL/pseudo-DBL and three BlaRs) among 813 BLDB sequences. When decreasing the identity threshold to 99%, 623 unique sequences were tagged with 653 reference sequences (650 DBL/pseudo-DBL and three BlaRs), while at 90%, 1,269 unique sequences were tagged with 801 reference sequences. All those tagged sequences are distributed across 18 clusters, regardless of the identity threshold. Interestingly, up to half of the reference sequences tag cluster 60 (i.e., 168 sequences at 100%; 363 at 99%; 452 at 90%). The main genus of this cluster composed of 66 representative sequences (standing for a total of 3,472 sequences) is *Acinetobacter*, which is the host organism for 99.7% of the sequences. Irrespective of the high-redundancy of cluster 60, the latter genus is known to harbor various chromosome-encoded DBL (10).

**Assessment of the beta-lactamase activity in uncharacterized clusters.** To test the beta-lactamase activity of some of the 46 non-annotated clusters, 10 DBL-homolog sequences were selected for expression and production. Clusters were sorted from the largest to the smallest (considering all and not only representative sequences), then one sequence from the first 10 clusters with no DBL found in the BLDB, a sequence length between 250 and 350 AAs and no mutation in the three conserved motifs defining the class D active site. Thus, the 10 DBL-homolog (termed OXAVL01 to 10) were selected from clusters 14, 22, 23, 28, 30, 39, 41, 42, 44, and 57 (Table S2). OXAVL01 has the two lysines of its active site mutated but these mutations are shared by all the sequences in cluster 14. According to the clinical score (see Prevalence of DBL-homolog genes in clinical strains), none of those DBL-homologs belong to a clinical strain (six classified as non-clinical and four as unclassified). Seven of those sequences are chromosome-encoded while no localization could be associated to OXAVL05, OXAVL09, and OXAVL10.

The OXAVL01-10 genes were cloned in the pET24a(+) plasmid under the control of the strong T7 promoter and introduced in *Escherichia coli*. The production of OXAVL01-10 was induced by IPTG and evaluated by SDS-PAGE and beta-lactam hydrolysis. No apparent over-expression of OXAVL01, OXAVL03, OXAVL05, OXAVL07, and OXAVL09 (OXA-1091) was observed in the soluble or insoluble fractions of *E. coli* (DE3) grown at 18°C and 37°C. For OXAVL04, OXAVL08, and OXAVL10, a large production of the beta-lactamases was found only in the insoluble fractions at both culture temperatures, likely indicating the formation of inclusion bodies. Only OXAVL02 (OXA-1089) and OXAVL06 (OXA-1090) were overproduced as soluble enzymes at 18°C.

**TABLE 3** Beta-lactamase activity of crude extract (CE) for cells expressing active DBL-homologs[a]

| CE | $V_0$ (nmol.min$^{-1}$.mgP$^{-1}$) | | | |
|---|---|---|---|---|
| | Nitrocefin | Ampicillin | Oxacillin | Imipenem |
| OXAVL02 | 9.5 | 70 | 18 | 4 |
| OXAVL03 | 1 | 1 | NH | NH |
| OXAVL04 | 0.7 | NH | NH | NH |
| OXAVL05 | 2. | NH | NH | NH |
| OXAVL06 | 6 | 85 | 100 | 7 |
| OXAVL09 | 4 | 7 | 4 | NH |
| OXAVL10 | 0.5 | NH | NH | NH |

[a]The measurements were performed in 25 mM HEPES buffer (pH 7) at 30°C. NH, no hydrolysis.

The evaluation of the beta-lactamase activity on crude cell extracts (Table 3) showed that only OXAVL02 and OXAVL06 were able to hydrolyze all beta-lactams tested, including imipenem. OXAVL09 was active versus nitrocefin, ampicillin, and oxacillin but not imipenem. OXAVL03 was able to hydrolyze nitrocefin and ampicillin. Cell extracts of OXAVL04, OXAVL05, and OXAVL10 were active only against nitrocefin. These results may only be indicative of the true spectrum of activity because of the low fraction of soluble enzymes present in some cases. The DBL-homolog enzymes were not produced in an active form in the strains bearing the plasmid pOXAVL01, pOXAVL07, or pOXAVL08.

**OXAVL02 and OXAVL06 have carbapenemase activity.** Because crude extracts of OXAVL02 and OXAVL06 were the only ones able to hydrolyze all tested beta-lactams and had the highest level of expression in the soluble fraction, we focused our work on those two hydrolases. The purification of the two enzymes included three chromatographic steps, namely, an anion exchanger, an IMAC affinity chromatography, and a molecular sieve. For OXAVL02, the purification consists in an IMAC column followed by a strong anion exchanger high resolution SOURCE 15Q column. The last step is a size exclusion chromatography (SEC). At the end of the process, we obtained more than 100 mg of pure protein per liter of culture. The three steps of the OXAVL06 purification are a Q Sepharose HP ion exchanger, an IMAC column, and finally a SEC. For OXAVL06, we obtained 10 mg of pure protein per liter of culture.

SEC experiments revealed that the OXAVL02 elutes in three major peaks (Fig. 3a), with one at an elution volume typical of a monomeric DBL (~260 mL). The two additional peaks elute at about 230 mL and 180 mL, which is similar to the elution volume of the dimer and multimer, respectively. The three peaks displayed an oxacillinase activity. Due to the low precision of oligomeric states of the proteins determined by SEC, we further characterized these three peaks using size exclusion chromatography coupled to multi-angle light scattering (SEC-MALS) (Fig. 3b).

The elution was monitored by a UV detector, a MALS detector, and a differential refractometer in line with the SEC column, allowing for the deconvolution of the protein molar masses (MM) of eluting protein complexes. The major peak in the OXAVL02 sample was confirmed to result from an equilibrium between a major monomeric form with an apparent protein MM of 32,000 ± 1,000 Da (theoretical MM [tMM] 31,298 Da) and a dimer at 62,000 ± 2,000 Da (tMM 62,596 Da). Of the two other peaks, the lower elution volume peak (at 180 mL) contained large aggregates (apparent MM > 3 × 10$^5$ Da), while the higher elution volume peak (230 mL) corresponded to the approximate MM of a dimer at 62,000 ± 2,000 Da (tMM 62,596 Da) in equilibrium with protein aggregates. Similar data were recorded for OXAVL06 (Fig. S2).

A kinetic profile of the two purified DBL-homologs was performed in the presence of 50 mM NaHCO$_3$ (Table 4). Indeed, in the absence of hydrogenocarbonate, their activity generally showed an initial burst, followed by a pronounced slowdown, even when the substrate conversion and product accumulation were quite low. Our data indicates that OXAVL02 displays a lower catalytic efficiency compared to OXAVL06. We observed that both enzymes were not able to hydrolyze amoxicillin, temocillin, cefazolin, and cefotaxime. In addition,

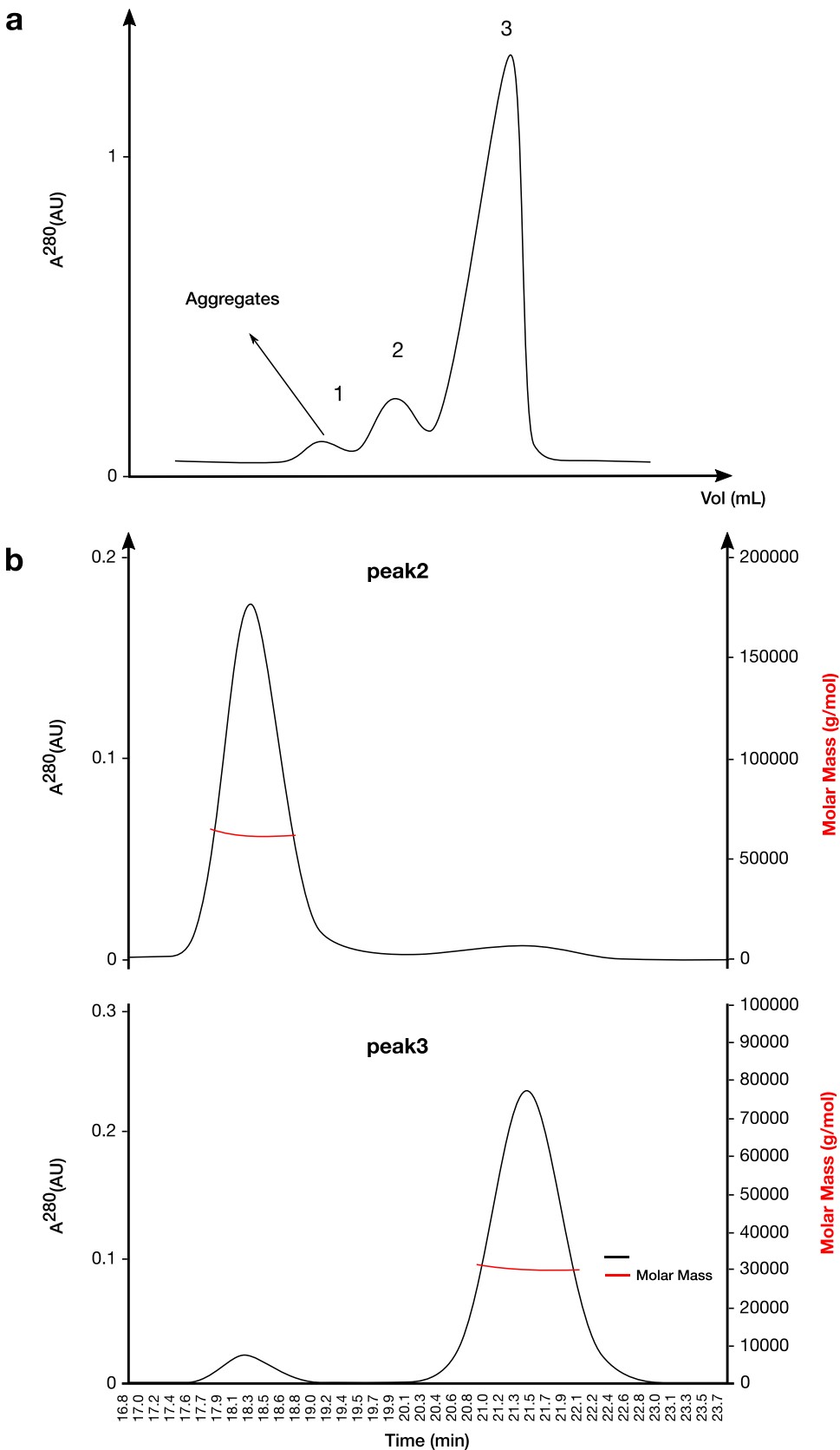

**FIG 3** SEC and SEC-MALS analysis performed on the purified OXAVL02. (a) SEC analysis of the purified OXAVL02. (b) Determination of the multimeric state of OXAVL02 (peaks 2 and 3) by SEC-MALS analysis.

**TABLE 4** Kinetic parameters of OXAVL02 and OXAVL06 beta-lactamases in 25 mM HEPES pH 7.5 + 50 mM NaCarbonate[a]

| Antibiotics | OXAVL02 | | | OXAVL06 | | |
|---|---|---|---|---|---|---|
| | $K_{cat}$ ($s^{-1}$) | Km ($\mu M$) | $K_{cat}$/Km ($\mu M^{-1}s^{-1}$) | $K_{cat}$ ($s^{-1}$) | Km ($\mu M$) | $K_{cat}$/Km ($\mu M^{-1}s^{-1}$) |
| Ampicillin | 20 ± 2 | 380 ± 10 | 0.055 ± 0.007 | 530 ± 30 | 270 ± 20 | 2 ± 0.3 |
| Carbenicillin | 22 ± 1 | 400 ± 20 | 0.055 ± 0.005 | 380 ± 20 | 1400 ± 200 | 0.25 ± 0.05 |
| Piperacillin | 3 ± 0.02 | 850 ± 30 | 0.0055 ± 0.0003 | NH | NH | NH |
| Oxacillin | 1 ± 0.05 | 690 ± 40 | 0.0015 ± 0.0002 | 90 ± 5 | 160 ± 20 | 0.56 ± 0.10 |
| Cephaloridine | >12.5 | >400 | 0.030 ± 0.005 | 24 ± 3 | 90 ± 10 | 0.26 ± 0.06 |
| Nitrocefin | Product Inhibition | | | 350 ± 30 | 20 ± 0.5 | 17.5 ± 2 |
| Imipenem | 9 ± 1 | 550 ± 50 | 0.016 ± 0.002 | 0.9 ± 0.05 | 0.4 ± 0.05 | 2.5 ± 0.03 |
| Meropenem | 0.3 ± 0.05 | 6 ± 0.2 | 0.05 ± 0.01 | NH | NH | NH |

[a]NH, no hydrolysis. Each kinetic value is the mean and standard deviation of three different measurements.

OXAVL06 was not active against piperacillin and meropenem. We confirmed also that the two beta-lactamases displayed a carbapenemase activity. Imipenem was among the best substrates ($k_{cat}/K_m$ = 0.016 and 2.5 $\mu M^{-1}s^{-1}$ for OXAVL02 and OXAVL06, respectively). In comparison to values obtained for oxacillin, the $k_{cat}/K_m$ ratios of OXAVL02 for meropenem and imipenem were 30- and 10-fold higher, respectively.

## DISCUSSION

**No OXA-domain family protein detected in Archaea.** The focus of this study was to explore the occurrence of class D beta-lactamases in the prokaryotic world. The 24,916 identified OXA-domain family sequences correspond to 3,510 unique sequences distributed across 20,343 bacterial strains. This highlighted a well-known redundancy in the NCBI RefSeq database toward clinical strains (26) (Fig. S3). The fact that none of these OXA-domain family proteins was detected in Archaea could be expected because Archaea are naturally resistant to beta-lactam antibiotics. Indeed, even when a pseudomurein is present, the cross-linking of the glycan chains does not involve d-Ala-d-Ala and thus does not hinge on the activity of penicillin binding proteins. However, two recent studies identified class A, B, and C beta-lactamase homologues in archaeal genomes and revealed that archaeal class B and C homologues do show a weak beta-lactamase activity (21, 28). Therefore, although we did not detect OXA-domain family proteins in Archaea, it is possible that archaeal OXA-domain family proteins will be identified in further studies, like for the other classes of beta-lactamases.

**Contaminated genomes from local NCBI RefSeq database.** Identification of new beta-lactamases in some unexpected organisms like Archaea or non-clinical bacterial strains might seem an exciting finding but could also be artifacts. In 2021, Lupo et al. assessed the contamination level of 111,088 bacterial genomes in the NCBI RefSeq database and found that 1% of the genomes were contaminated at a minimal threshold of 5% (26). For the 20,343 genome assemblies used in the current study, 20,200 results were available, indicating that 143 genomes had been suppressed since then. Among these 20,200 bacterial genomes, 114 showed a contamination level ≥5%. Those 114 genomes are distributed across seven phyla: Proteobacteria (78), Firmicutes (29), Verrucomicrobia (2), Cyanobacteria (2), Chloroflexi (1), Chlorobi (1), and Balneolaeota (1). Obviously, conclusions for contaminated genomes should be taken with caution. For example, the only genome containing a DBL-homolog sequence in the Balneolaeota phylum is contaminated. From our data, it is however difficult to identify if this DBL-homolog is part of the contamination or if it is genuinely part of the genome, possibly acquired from an unknown organism by horizontal gene transfer.

**OXA-domain family phylogeny and classification.** We inferred the phylogenetic tree using a matrix of the 188 most conserved AAs (around two thirds of typical DBL length) from the 1,413 representative OXA-domain family sequences. Those representative sequences resulted from the deduplication of the 3,510 unique sequences at a global identity threshold of 95%, which means that, considering their full length, they are similar to at least 95% of observed identity with member sequences of their deduplication clusters. Then, a phylogenetic clustering of the representative OXA-domain family

proteins was computed using the patristic distances taken from the tree (i.e., the sum of the branch lengths between two leaves). This patristic distance quantifies the number of AA substitutions computed by the statistical model of sequence evolution. To select the best clustering parameters, we decided to exclude the clustering solutions with less than 15 clusters. In fact, we noticed that at least half of the OXA-domain family protein sequences regroup into one single large cluster when fewer than 15 clusters are produced. The retained parameters yielded 64 clusters, including 19 of size one (singletons). Despite a larger number of clusters, BLDB reference DBL and pseudo-DBL sequences are distributed in only 16 clusters, while BlaR from *Clostridium difficile* and *Bacillus licheniformis* are found in cluster 15, and BlaR from *Staphylococcus aureus* in cluster 18.

Another objective of this study was to suggest a meaningful classification of OXA-domain family proteins based on their phylogeny. However, the majority of bacterial strains have only one OXA-domain family protein, indicating that these genes are essentially orthologous. Moreover, 50 clusters out of 64 contain sequences from organisms belonging to a single phylum or class, which means that sequence diversity within the OXA-domain family is mostly due to speciation. While it is possible to generate a classification of class D beta-lactamases based on the clustering obtained in this study, our results indicate that a more practical classification should rather include a reference to the species of origin.

**Analysis of the BlaR clusters.** Because some class D beta-lactamases display more sequence identity with the C-terminal beta-lactam sensing domain of BlaR than with other class D beta-lactamases, it was impossible to avoid retrieving types of proteins in our homology searches. BlaR is also characterized by a N-terminal domain containing four transmembrane helices and a zinc protease module in loop 2 that is activated upon acylation of the C-terminal domain catalytic serine by a beta-lactam antibiotic. This triggers a cascade that eventually leads to the increased expression of either a beta-lactamase or a resistant PBP (19). As a consequence, BlaR has a total length of about 600 AAs. The size was therefore used to discriminate between BlaR-homologs (>550 AAs) and the DBL-homologs (<350 AAs). Clusters 16 and 17 exclusively contain BlaR-homologs, while clusters 8, 15, and 18 contain both BlaR- and DBL-homologs (Table S1). Most BlaR-homologs harbor a polar residue as the third residue of the second conserved motif (Table S1) and contain a N-terminal peptidase domain, two specific features of the BlaR receptor. The only exceptions are a few shorter sequences found in cluster 18; which have been removed from the database since we downloaded them, possibly indicating sequencing errors. Two sequences shorter than 550 AAs and labeled as BlaR-homologs are found in cluster 15. They have the typical conserved motifs of BlaR but their N-terminal domain is truncated and likely not functional. In this study we noticed that BlaR-homolog genes are more frequent on a plasmid nearby a transposase gene. A recent study has shown that *Staphylococcus* species have Tn552-like elements carrying the *bla* operon often located on a plasmid (29). The authors hypothesized that the Tn552 transposon can mediate the transfer of the *bla* operon from a plasmid to the chromosome. This hypothesis would also fit our results showing a high prevalence of BlaR-homolog genes on plasmids and their proximity with transposase genes.

**Analysis of all the 62 DBL-homolog clusters.** The size of the DBL-homolog proteins is very homogenous and the only sequences longer than 350 AAs are three fusions between a class D and a class C beta-lactamase (cluster 8), possibly homologous to LRA-13 (20), three sequences with an N-terminal extension (up to 423 AAs in total) in cluster 21, and a fusion with a crotonase domain (one sequence in cluster 40) of unknown function. The analysis of the active site motifs (Table S1) shows an almost perfect conservation of the three motifs characteristic of the catalytic site (SxxK, SxV, and KTG), as well as of the tryptophan in the omega loop, which is important for the stabilization of the carboxylated lysine of the first motif. The most variable position is the second motif valine, which is often substituted by another hydrophobic AA. Some clusters do however diverge from this consensus. Indeed, clusters 13, 29, 35, 36, 45, and 54 contain only one or two sequences and have significantly impaired motifs,

which are likely not compatible with a beta-lactamase activity. Cluster 12 (eight representative sequences), which also has poorly conserved motifs, with an absence of catalytic serine in most cases, is very unlikely to display beta-lactamase activity. In contrast, cluster 14 (11 representative sequences) has the following conserved motifs: SxxH, SxH/Q, AS/TG. A sequence from this cluster (OXAVL01) was selected for *in vitro* characterization. No beta-lactamase activity was measured on a crude extract, but no overexpression was detected in our assays, preventing us from drawing any definitive conclusion. The conserved motifs are, however, not sufficient to warrant a beta-lactamase activity, as demonstrated by cluster 19 (Table S1), which regroups so far only pseudo-DBL sequences like YbxI or BAC-1 (17, 18).

**Probing clusters without class D beta-lactamase representative.** Beyond OXAVL01, we have selected nine DBL-homologs among the 45 clusters devoid of reference OXA-domain family proteins to probe their activity. Overall, for seven of the 10 sequences selected for evaluation, a beta-lactamase activity was detected at least on crude extracts (Table 3), including two hydrolases active on imipenem (OXAVL02 and OXAVL06). The enzymatic studies of these two DBL-homolog enzymes confirmed that they both display a beta-lactamase activity and hydrolyze efficiently imipenem but that meropenem is only inactivated by OXAVL02. We also showed that the presence of hydrogenocarbonate enhances their catalytic activity, a sign of the necessary carboxylation of the first motif lysine for optimal activity. As already shown for numerous other class D beta-lactamases, the monomeric form OXAVL02 is in equilibrium with the dimeric form, the monomer being the predominant form of the enzyme at the concentration tested. These results, obtained with randomly selected enzymes, confirm that the environmental strains provide a large reservoir of new resistance genes, which include high potential for resistance to carbapenems, a family of last resort antibacterials. The acquisition of such genes by multi-resistant nosocomial strains therefore represents an important threat for the treatment of the related infections. This phenomenon has already been observed with the chromosome-encoded class A CTX-M-3 from *Kluyvera* spp., which is at the origin of the plasmid-borne CTX-M-1-derived cefotaximases produced by clinical isolates (30). This is a reminder of the importance of an adequate use of the available antibiotics to postpone as much as possible the emergence of new resistance factors.

**Predicting activity profiles from amino acid sequences.** The most clinically relevant result would be to deduce the activity profile of an enzyme from its AA sequence. However, determining the activity of only one representative DBL-homolog per cluster would not be informative of the specific activity profile of the cluster. In fact, it has been shown that only one mutated AA can alter the activity profile of a DBL (8, 9). Although the sequence similarity between the 1,413 representative sequences and their respective member sequences is high (i.e., at least 95% identity), the identity between the sequences within one of the 45 non-singleton phylogenetic cluster is low (i.e., down to 50%) (Table S3). Furthermore, this similarity is certainly undervalued because it is computed from only 188 unambiguously aligned AAs. Altogether, those arguments support that, for now, the activity profile of a DBL-homolog cannot be predicted only based on its AA sequence. This problem is also true for the other classes of beta-lactamases. Solving this would require a major effort for the high throughput biochemical characterization of the enzymes and the determination of their three-dimensional structure, which is more likely correlated with the substrate specificity than the AA sequence. While biochemical characterization still represents a significant bottleneck, the recent development of the AlphaFold prediction software (31) has put structure determination within reach. Consequently, the use of artificial intelligence to predict the activity profile of enzymes is not as far-fetched as it used to be.

## MATERIALS AND METHODS

**SQL database.** Bioinformatic data generated in this study were stored into a sqlite3 database (Fig. S1). This database was exploited using SQL queries in order to generate additional results and statistics.

**Reference class D beta-lactamase sequences and identification of OXA-domain family proteins.** A total of 1,617 unique beta-lactamase amino-acid sequences were downloaded from the NCBI

Pathogen Detection server (ftp://ftp.ncbi.nlm.nih.gov/pathogen/) on December 1, 2017. Among those, 470 DBL were retrieved based on metadata and accession numbers. DBL protein sequences were deduplicated using CD-HIT v4.6 (32) with a global sequence identity threshold of 0.98 and then aligned using MAFFT v7.273 (33). An HMM profile was constructed from the DBL alignment using the HMMER package v3.1b2 (34) to identify OXA-domain family proteins in a local prokaryotic protein sequence database. This local database was built on December 7, 2017 using the protein sequences of 80,490 prokaryotic genome assemblies stored in the NCBI RefSeq database. OXA-domain family proteins were graphically selected using the ompa-pa.pl interactive software package (A. Bertrand and D. Baurain; https://metacpan.org/dist/Bio-MUST-Apps-OmpaPa) and taxonomically annotated using the NCBI Taxonomy.

**Annotation of OXA-domain family proteins.** OXA-domain family proteins were tagged using a BLAST-based annotation script (part of Bio-MUST-Drivers) with an identity threshold from 90% to 100% and an e-value threshold of 1e-20. DBL-homolog sequences used for the annotation were downloaded from the BLDB (http://www.bldb.eu/BLDB.php?prot=D) (11) on July 22, 2019, to which were added three sequences of the membrane receptor BlaR from *Clostridium difficile* (CDT53463.1), *Staphylococcus aureus* (P18357), and *Bacillus licheniformis* (P12287), the bifunctional class C/class D beta-lactamase LRA13-1 (ACH58991.1) (20) and the two intrinsic pseudo-DBLs of *Clostridium difficile* CDD-1 (CZR76508.1) and CDD-2 (SJQ22628.1) (27).

**Domain characterization of OXA-domain family proteins.** The potential presence of a signal peptide was predicted using local SignalP-5.0b (35). The organism option was set to "Gram+" for sequences belonging to Firmicutes and Actinobacteria and "Gram-" for the other phyla. To improve the prediction of transmembrane helices with local TMHMM v2.0 (36), the signal peptide (if any) was first removed from the original sequences when the cleavage site prediction probability was greater than or equal to 0.6. For sequences of intermediary length (i.e., between 350 and 550 AAs) and some long sequences (i.e., greater than 550 AAs), InterProScan v5.37-76.0 with default parameters and disabled use of the precalculated match lookup (37), along with pepwindowall with default parameters from the EMBOSS web portal (38) were used to distinguish between transmembrane segments and other extensions.

**Localization and genetic environment of OXA-domain family proteins.** A genetic environment database was built from the bacterial genomes featuring at least one OXA-domain family sequence using GeneSpy "3 in 1" module, as described in the manual (39). Contig accessions were retrieved from the database and the corresponding FASTA files were downloaded using the command-line version of the "efetch" tool from the NCBI Entrez Programming Utilities (E-utilities). PlasFlow v1.1 was used to predict potential plasmid sequences in the contig FASTA files (40).

**Clinical strain determination.** BioSample reports associated with bacterial organisms were also downloaded using efetch (see above). All words of a report were collected and fed to a script that renamed and standardized them using an OBO (Open Biomedical Ontologies) dictionary. A score was attributed to each standardized word: +1 for a "clinical" word, 0 for an uninformative word and –1 for a non-clinical word. At last, a final score was computed for each BioSample according to its collection of standardized words (see figshare). A bacterial strain was considered as "clinical" when its metadata were associated with a positive score, "non-clinical" for a negative score and not classified for a null score.

**Alignment and phylogenetic analysis.** After deduplication using CD-HIT v4.6 (32) with a global sequence identity threshold of 0.95, OXA-domain family protein sequences were aligned using MAFFT v7.273 (33). Alignments were then carefully optimized by hand using the program "ed" and alignment columns were manually selected using the program "net," both part of the MUST software package (41). The resulting matrix of 1,413 sequences × 188 unambiguously aligned AAs was used to infer a phylogenetic tree with RAxML v8.1.17 (42) under the LG+F+G4 model. Support values were initially estimated through 100 fast bootstrap pseudo-replicates with RAxML then transformed into transfer bootstrap expectation (TBE) values using the booster algorithm (43).

**Phylogenetic clustering.** To produce clusters of related OXA-domain family sequences, the phylogenetic tree was first converted to a phylo4 object using the readNewick function of the phylobase R package (44). Then, a patristic distance matrix (dist.mat) was computed using the distTip function from the adephylo R package (45) and an adjacency matrix (adj.mat) was computed as follows: $ajd.mat = \frac{dist.mat}{lim.p} < 1$ with $lim.p = max(dist.mat) \times x$ and $x$ varying from 0.10 to 0.50. Clustering was performed by passing the adjacency matrix to the mcl function of the MCL R package (46) with the addLoops option set to FALSE, allow1 set to TRUE and the inflation parameter varying from 1.0 to 3.0 by increments of 0.5. The best parameter combination was chosen by maximizing a score composed of the normalized entropy and the fraction of monophyletic clusters, following the method of Califice et al. (47). However, combinations yielding less than 15 clusters were discarded, regardless of their score, in order to avoid the grouping of most OXA-domain family sequences into a single cluster and retain the potential to provide a meaningful classification.

**DBL-homolog genes selection for lab validation.** Based on the phylogenetic clustering of OXA-domain family proteins, 10 representative protein sequences (hereafter referred to as OXAVL01 to OXAVL10) spread among different clusters corresponding to DBL-homologs were selected as probes for the functional diversity. The criteria of selection were: (i) the sequence must belong to a cluster with more than five DBL-homologs and no DBL found in the BLDB; (ii) the length of the sequence must lie between 250 and 350 AAs and the sequence must have no mutation in the three conserved motifs defining the class D active site (SxxK, SxV, KT/SG) (except if a mutation is shared by all the sequences of the cluster); (iii) the sequence must be present in a bacterial species where no DBL is described according to the BLDB.

**Gene synthesis and expression plasmids.** Signal peptides of the 10 selected sequences were removed and replaced by the PelB leader sequence (48). Then, the corresponding genes were

synthesized after codon optimization for expression in *E. coli*. Expression plasmids of OXAVL01 to OXAVL10 were purchased from Twist Bioscience (San Francisco, CA, USA). Briefly, the synthesized genes were cloned into pET24a(+) (Novagen-Merck KGaA, Darmstadt, DE) and inserted between BamHI (at the 5′ end of the gene) and XhoI (at 3′ end of the gene) restriction sites. All the enzymes were produced by *E. coli* BL21(DE3) (Fisher Scientific SAS Illkirch Cedex, FR) carrying pOXAVL01-pOXAVL10 plasmids in LB medium supplemented with kanamycin 50 $\mu$g/mL (LB-kanamycin).

**Antibiotics.** Kanamycin was purchased from MP Biomedicals; cefotaxime, cephaloridine, and oxacillin from Sigma-Aldrich; cefazolin from Pharmacia & Upjohn SpA; imipenem from MSD; meropenem from Fresenius Kabi NV/SA; ampicillin from Fisher Scientific; amoxicillin from PanPharma; carbenicillin from Pfizer Italy; piperacillin from Lederle/AHP Pharma; temocillin from Eumedica N.V/S.A; and nitrocefin from Abcam.

**Assessment of soluble enzymes expression levels.** Six mL of LB-kanamycin was inoculated with single colonies of *E. coli* BL21(DE3) carrying the plasmids pOXAVL01 to pOXAVL10. The precultures were incubated overnight (O/N) at 37°C with orbital shaking at 250 rpm. Next, 2.5 mL of the different precultures were added to 100 mL of fresh LB-kanamycin. The bacteria were grown to an $A_{600}$ of 0.7 and IPTG was added at a final concentration of 0.5 mM. The different cultures were divided in two, one incubated at 37°C and the other one at 18°C. Aliquots (1 mL) of the different cultures at 37°C were taken 0 h, 2 h, and 4 h after induction. In the case of the cultures incubated at 18°C, two aliquots (0 h and 24 h after induction) were analyzed. The different aliquots were centrifuged at 5,000 *g* for 10 min, the bacterial pellets were resuspended in 25 mM HEPES buffer (pH 7.0) and sonicated (three times for 30 seconds each time at 12 watts [W]). Cell debris was eliminated by centrifugation at 13,000 *g* for 30 min. 20 $\mu$L of the soluble fractions and pellets were loaded onto a sodium dodecyl sulfate polyacrylamide gel (SDS-PAGE) (4-20%). The run was performed at a constant voltage (120 V). The beta-lactamase activity of the different fractions was determined by measuring the initial rate of hydrolysis of 100 $\mu$M Nitrocefin, 1 mM oxacillin, 1 mM ampicillin, and 100 $\mu$M imipenem.

**Production and purification of OXAVL02 and OXAVL06.** One hundred mL of LB-kanamycin was inoculated with a single colony of *E. coli* BL21(DE3) pOXAVL02 or *E. coli* BL21(DE3) pOXAVL06. The preculture was incubated O/N at 37°C under agitation. Then, 40 mL of the preculture was added to 1 L of fresh LB-kanamycin. IPTG (100 $\mu$M final concentration) was added when the culture reached an $A_{600}$ of 0.7. The cultures were incubated O/N at 18°C. Cells were harvested by centrifugation at 5,000 *g* for 10 min at 4°C. The pellets were resuspended in 15 mL 50 mM Sodium Phosphate, 0.5 M NaCl, 20 mM Imidazole pH 8.0 (buffer A) for pOXAVL02, and in 25 mM HEPES pH 7.0 (buffer B) for pOXAVL06. The bacteria were disrupted with a cell disrupter (Emulsiflex C3 Avestin GmbH, DE), which allows cell lysis at a pressure of 5,500 kPa. The lysates were isolated by centrifugation at 45,000 *g* for 30 min. The two supernatants were dialyzed O/N at 4°C against buffers A and B, respectively. The dialyzes samples were then filtered through a 0.45 $\mu$m filter.

For OXAVL02, the supernatant was loaded onto Ni Sepharose (24 mL) (GE Healthcare Europe GmbH, Freiburg) previously equilibrated with buffer A. The enzymes were eluted with a gradient using 50 mM Sodium Phosphate pH 8.0, 0.5 M NaCl, 0.5 M imidazole (buffer C). The fractions displaying a beta-lactamase activity were pooled, and then dialyzed O/N against buffer B and loaded onto a Source 15 Q column 20 mL (Pharmacia Biotech/BioSurplus Inc., San Diego, CA, USA) equilibrated with the same buffer. The enzyme was eluted with a salt gradient using buffer B with 1 M NaCl (buffer D). The fractions were pooled and loaded on a molecular sieve Superdex 75 GL 500 mL column (GE Healthcare Europe GmbH, Freiburg, DE) equilibrated in buffer B.

Because the production level of OXAVL06 was much lower, the first two purification steps were inverted compared with OXAVL02. This strategy avoided a poor efficiency of the Ni Sepharose column caused by an unspecific binding of the crude protein extract that saturates the matrix. Hence, the cleared supernatant was loaded onto a 10 mL Q Sepharose HP column (GE Healthcare Europe GmbH, Freiburg) equilibrated in buffer B. The enzyme was eluted with a salt gradient using buffer D. The fractions with a beta-lactamase activity were pooled, and dialyzed O/N in 50 mM Sodium Phosphate pH 7.5, 0.5 M NaCl, 20 mM imidazole (buffer E). The dialyzes sample was loaded onto Ni Sepharose (24 mL) (GE Healthcare Europe GmbH, Freiburg) previously equilibrated with buffer E. The enzymes were eluted with a gradient using 50 mM Sodium Phosphate pH 7.5, 0.5M NaCl, 0.5 M imidazole pH 7.5 . The active fractions were collected and concentrated by ultrafiltration on a YM-10 membrane (Amicon) to a final volume of 2 mL, then loaded onto a molecular sieve Superdex 75 GL (10/300) column (GE Healthcare Europe GmbH, Freiburg) equilibrated in buffer B.

**Conformational characterization of OXAVL02 and OXAVL06.** The oligomeric states of the DBL-homolog enzymes were analyzed by SEC-MALS (Treos II, WYATT Technology France) (49). The experiments were performed using a HPLC Bio-inert Shimadzu Prominence LC-20Ai (SHIMADZU Benelux B.V) coupled to a SPD-20A UV/VIS detector and a RID-20 refractive index detector. The different active fractions isolated by size exclusion chromatography were dialyzed against a "SECMALS-PBS buffer" ($Na_2HPO_4$ 10 mM, $KH_2PO_4$ 1.8 mM, NaCl 137 mM, KCl 2.7 mM pH 7.4). Samples (100 $\mu$L OXAVL02 or OXAVL06 at 0.5 to 1 mg/mL) were loaded onto a Superdex 200 Increase 10/300 G column (GE Healthcare Bio-Sciences AB Uppsala) pre-equilibrated with the "SECMALS-PBS buffer." The column was calibrated by using bovine serum albumin (BSA) (MM = 66,430 Da) as reference standard. The data acquisition of molecular mass, distribution of Monomer-Dimer equilibrium, and percentage of aggregates were estimated using the ASTRA software (49).

**Kinetic constants determination.** Steady-state kinetic constants ($K_m$ and $k_{cat}$) were determined by measuring substrate hydrolysis under initial rate conditions and using the Hanes–Woolf linearization of the Michaelis–Menten equation (50). Kinetic experiments were performed by following the hydrolysis of each substrate at 30°C in 50 mM HEPES buffer pH 7.5, 50 mM $Na_2CO_3$. The reactions were performed in a total volume of 500 $\mu$L at 30°C. BSA (20 $\mu$g/mL) was added to diluted solutions of beta-lactamase in

order to prevent enzyme denaturation. The data were collected with a Specord 50 PLUS spectrophotometer (Analytik Jena). Each kinetic value is the mean of three different measurements.

**Data availability.** Publicly available data sets analyzed in this study and the companion SQL database can be found here: https://doi.org/10.6084/m9.figshare.18544955.

## SUPPLEMENTAL MATERIAL

Supplemental material is available online only.

**SUPPLEMENTAL FILE 1**, PDF file, 2.4 MB.

## ACKNOWLEDGMENTS

V.L. is supported by a Fonds pour la Formation à la Recherche dans l'Industrie et dans l'Agriculture (FRIA) fellowship of the FRS-FNRS. F.K. is a research associate of the FRS-FNRS. M.G. and P.S.M. were supported by the Belgian Federal Public Service Health, Food Chain Safety and Environment (Grant No. RF 17/6317 RU-BLA-ESBL-CPE). Computational resources were provided through two grants to DB (University of Liège "Crédit de démarrage 2012" SFRD-12/04; FRS-FNRS "Crédit de recherche 2014" CDR J.0080.15). We thank Mohammed Terrak and Adrien Boes for useful advice at the initial stage of the project and Hiba Jabri for her help with the OBO dictionary.

V.L. performed experiments, analyzed the data, drew the figures, wrote the manuscript and approved the final manuscript. D.B. and F.K. conceived the study and designed experiments, analyzed the data, wrote and reviewed the manuscript and approved the final manuscript. J-M.F. conceived the study, wrote and reviewed the manuscript ,and approved the final manuscript. P.S.M. performed experiments, wrote the manuscript, and approved the final manuscript. M.G. designed experiments, wrote the manuscript, and approved the final manuscript. B.J. analyzed the data, reviewed the manuscript, and approved the final manuscript.

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
