## [Reviewer comments · Microbiology Spectrum]

Microbiology Spectrum

An extended reservoir of class-D beta-lactamases in non-clinical bacterial strains

Valérian Lupo, Paola Sandra Mercuri, Jean-Marie Frère, Bernard Joris, Moreno Galleni, Denis Baurain, and Frederic Kerff

Corresponding Author(s): Denis Baurain, University of Liège

Review Timeline:

Submission Date:	January 26, 2022
Editorial Decision:	February 10, 2022
Revision Received:	February 18, 2022
Accepted:	February 20, 2022

Editor: Monica Garcia-Solache

Reviewer(s): Disclosure of reviewer identity is with reference to reviewer comments included in decision letter(s). The following individuals involved in review of your submission have agreed to reveal their identity: Peter Oelschlaeger (Reviewer #1)

Transaction Report:

DOI: <https://doi.org/10.1128/spectrum.00315-22>

February 10, 2022

Prof. Denis Baurain
University of Liège
InBioS-PhytoSYSTEMS, Eukaryotic Phylogenomics
Quartier Vallée 1
chemin de la Vallée 4
Liège, Liège 4000
Belgium

Re: Spectrum00315-22 (An extended reservoir of class-D beta-lactamases in non-clinical bacterial strains)

Dear Prof. Denis Baurain:

Thank you for submitting your manuscript to Microbiology Spectrum. As you will see your paper is very close to acceptance. Please modify the manuscript along the lines the reviewers recommended. As these revisions are quite minor, I expect that you should be able to turn in the revised paper in less than 30 days, if not sooner. You will find the reviewers' comments below.

When submitting the revised version of your paper, please provide (1) point-by-point responses to the issues raised by the reviewers, and (2) a PDF file that indicates the changes from the original submission (by highlighting or underlining the changes) as file type "Marked Up Manuscript - For Review Only". Please use this link to submit your revised manuscript. Detailed instructions on submitting your revised paper are below.

Link Not Available

Sincerely,

Monica Garcia-Solache

Reviewer comments:

Reviewer #1 (Public repository details (Required)):

A SQL database was created for this study. It would be nice if it became publicly accessible upon publication.

Reviewer #1 (Comments for the Author):

This manuscript presents an effective combination of bioinformatics and experimental work to explore the reservoir of class D beta-lactamases in non-clinical bacterial strains. The bioinformatics portion identified thousands of class D beta-lactamase genes and BlaR membrane receptor genes from the NCBI Pathogen Detection server. Most of these genes were encoded on the chromosome, followed by plasmids and unclassified. Based on homology, the proteins were clustered into 64 clusters, a few of which contained the known OXA family enzymes, while others contained exclusively novel enzymes. None of these genes were found in Archaea. A large portion of the genes were found close to transposons, which could facilitate their transfer to pathogenic strains. The new bioinformatics data created in this study is being hosted in a SQL database. It remained unclear to this author whether and how this database will be accessible publicly.

Regarding experimental work, ten class D enzyme genes were selected from clusters that did not contain known enzyme genes and were expressed in *E. coli* and analyzed. Two of these enzymes were expressed in the soluble fraction of cell lysates and

could be purified and studied biochemically. These purified enzymes were able to inactivate a broad range of beta-lactams, including the carbapenem imipenem.

In summary, the study demonstrates that a large non-clinical reservoir of class D beta-lactamases and BlaR membrane receptor genes exists that could potentially be transferred into clinical strains and cause problems in the clinic.

In the following are line-by-line comments/suggestions.

Line 105: "define" instead of "design"?

Line 112: "resistance" instead of "resistant".

Line 172: replace "greater or equal to" with "greater than or equal to"

Lines 251-252: Please consider rewriting to "Six mL of LB medium supplemented with kanamycin was inoculated with single colonies ..." (accordingly lines 269-270)

Line 262: replace "were" with "was"

Line 268: replace "OXAVL02/06" with "OXAVL02 and OXAVL06" (accordingly line 303)

Lines 274-276: It might be helpful to explain why different buffers and a different sequence of purification steps were used for the two enzymes.

Line 303: Since SEC-MALS is mentioned for the first time here, it would be good to spell out what the acronym means.

Line 317 and following: Please consider replacing "parameters" with "constants".

Line 326: It is amazing to have such small standard deviations.

Line 428: Replace "an" with "a"

Line 467: Some explanation as to what "CDD enzymes" are might be warranted. Cytidine deaminases?

Line 504: A verb is missing. Perhaps "..., of which 34% were DBL-homolog sequences."?

Line 539: Replace "cluster" with "clusters"

Line 547: Consider replacing "unknown" with "unclassified" for consistency with Table S2.

Lines 549-562: Perhaps state the conditions here again (although they are in the Materials & Methods): pET24a(+)-based plasmid with T7 promoter with induction (0.5 mM IPTG). This is important, because it represents an unnatural overproduction condition, which could be responsible for proteins being expressed in the insoluble form (inclusion bodies?). Were any attempts made to refold those proteins? The residual activity of all enzymes listed in Table 3 against nitrocefin (the best substrates of the four tested) can probably be explained with a small portion of the proteins going into solution.

Line 731: change "represent" to "represents"

Line 922: replace "broken" with "broken down"

Reviewer #2 (Public repository details (Required)):

Data are already publicly available. A link is provided in the Data Availability section of paper (lines 327 - 329).
<https://doi.org/10.6084/m9.figshare.18544955>

Reviewer #2 (Comments for the Author):

I found this study and paper to be technically sound, with enough detail to reproduce the study. It is an interesting exploration of Class D beta-lactamases and their hosts. In the attached review are grammatical suggestions for clarification.

Preparing Revision Guidelines

- point-by-point responses to the issues I raised in your cover letter
- Upload a compare copy of the manuscript (without figures) as a "Marked-Up Manuscript" file.
- Each figure must be uploaded as a separate file, and any multipanel figures must be assembled into one file.
- Manuscript: A .DOC version of the revised manuscript
- Figures: Editable, high-resolution, individual figure files are required at revision, TIFF or EPS files are preferred

Please return the manuscript within 60 days; if you cannot complete the modification within this time period, please contact me. If you do not wish to modify the manuscript and prefer to submit it to another journal, please notify me of your decision immediately so that the manuscript may be formally withdrawn from consideration by Microbiology Spectrum.

1 An extended reservoir of class-D beta- 2 lactamases in non-clinical bacterial 3 strains

Valérian Lupo ^{1,2}, Paola Sandra Mercuri ², Jean-Marie Frère ², Bernard Joris ²,
Moreno Galleni ², Denis Baurain ^{1,#}, Frédéric Kerff ^{2,#}

[revised manuscript text omitted]

Antibiotics	OXAVL02			OXAVL06		
	k _{cat} (s ⁻¹)	K _m (μM)	k _{cat} /K _m (μM ⁻¹ s ⁻¹)	k _{cat} (s ⁻¹)	K _m (μM)	k _{cat} /K _m (μM ⁻¹ s ⁻¹)
Ampicillin	20	380	0.055	530	270	2
Carbenicillin	22	400	0.055	380	1400	0.25

Piperacillin	3	850	0.0055	NH	NH	NH
Oxacillin	1	690	0.0015	90	160	0.56
Cephaloridine	>12.5	>400	0.030	24	90	0.26
Nitrocefin	Product Inhibition			350	20	17.5
Imipenem	9	550	0.016	0.9	0.4	2.5
Meropenem	0.3	6	0.05	NH	NH	NH

 **Figure 1. Classification of OXA-domain family protein sequences as DBL-**
 **homologs or BlaR-homologs. (a)** Length distribution of the 24,916 OXA-domain
 family protein sequences. Sequences shorter than 350 AAs are colored in blue,
 sequences longer than 550 AAs are in orange, while sequences between 350 and
 550 AAs are in red. **(b)** Length distribution of the representative sequences mapped

onto the phylogenetic tree. The tree was constructed from a matrix of 1413
 representative sequences x 188 unambiguously aligned AAs using RAxML under the
 LG+F+G4 model. (c) Distribution of the number of DBL-homolog and BlaR-homolog
 sequences per organism. Blue bar plots represent DBL-homolog sequences while
 orange bar plots represent BlaR-homolog sequences. The Y axis is in log10 units.

 **Figure 2. DBL-homolog and BlaR-homolog genes in their genetic context. (a)**
 Distribution of DBL- and BlaR-homolog genes according to the type of encoding
 molecule. (b) Distribution of the distances between DBL- and BlaR-homolog genes
 and transposase genes across the classified contigs. The distance is measured as a
 range of genes centered on the gene (DBL- or BlaR-homolog) of interest. DBL- and
 BlaR-homolog genes are colored in blue and orange, respectively.

Figure 3. **SEC and SEC-MALS analysis performed on the purified OXAVL02.** (a)

SEC analysis of the purified OXAVL02. (b) Determination of the multimeric state of

OXAVL02 (peaks 2 and 3) by SEC-MALS analysis.

Reviewer 1

A SQL database was created for this study. It would be nice if it became publicly accessible upon publication.

>>> This SQL database is publicly accessible through the link provided in the "Data Availability" section. However, the following sentence was added to the latter section to make it clearer:

Publicly available datasets analyzed in this study and the companion SQL database can be found here: <https://doi.org/10.6084/m9.figshare.18544955>.

This manuscript presents an effective combination of bioinformatics and experimental work to explore the reservoir of class D beta-lactamases in non-clinical bacterial strains. The bioinformatics portion identified thousands of class D beta-lactamase genes and BlaR membrane receptor genes from the NCBI Pathogen Detection server. Most of these genes were encoded on the chromosome, followed by plasmids and unclassified. Based on homology, the proteins were clustered into 64 clusters, a few of which contained the known OXA family enzymes, while others contained exclusively novel enzymes. None of these genes were found in Archaea. A large portion of the genes were found close to transposons, which could facilitate their transfer to pathogenic strains. The new bioinformatics data created in this study is being hosted in a SQL database. It remained unclear to this author whether and how this database will be accessible publicly.

Regarding experimental work, ten class D enzyme genes were selected from clusters that did not contain known enzyme genes and were expressed in *E. coli* and analyzed. Two of these enzymes were expressed in the soluble fraction of cell lysates and could be purified and studied biochemically. These purified enzymes were able to inactivate a broad range of beta-lactams, including the carbapenem imipenem.

In summary, the study demonstrates that a large non-clinical reservoir of class D beta-lactamases and BlaR membrane receptor genes exists that could potentially be transferred into clinical strains and cause problems in the clinic.

>>> Thank you for this accurate assessment of our work.

In the following are line-by-line comments/suggestions.

Line 105: "define" instead of "design"?

>>> Indeed, this word is more appropriate. Thank you.

Line 112: "resistance" instead of "resistant".

>>> Done. Thank you.

Line 172: replace "greater or equal to" with "greater than or equal to"

>>> Done. Thank you.

Lines 251-252: Please consider rewriting to "Six mL of LB medium supplemented with kanamycin was inoculated with single colonies ..." (accordingly lines 269-270)

>>> Done. Thank you.

Line 262: replace "were" with "was"

>>> Done. Thank you.

Line 268: replace "OXAVL02/06" with "OXAVL02 and OXAVL06" (accordingly line 303)

>>> Done. Thank you.

Lines 274-276: It might be helpful to explain why different buffers and a different sequence of purification steps were used for the two enzymes.

>>> The following sentences have been added:

Because the production level of OXAVL06 was much lower, the first two purification steps were inverted compared to OXAVL02. This strategy avoided a poor efficiency of the Ni Sepharose® column caused by an unspecific binding of the crude protein extract that saturates the matrix.

Line 303: Since SEC-MALS is mentioned for the first time here, it would be good to spell out what the acronym means.

>>> Done. Thank you.

Line 317 and following: Please consider replacing "parameters" with "constants".

>>> Done. Thank you.

Line 326: It is amazing to have such small standard deviations.

>>> Thank you for having brought this to our attention. The actual standard deviation values have now been added in Table 4 for full transparency.

Line 428: Replace "an" with "a"

>>> Done. Thank you.

Line 467: Some explanation as to what "CDD enzymes" are might be warranted. Cytidine deaminases?

>>> These CDD enzymes correspond to pseudo-class D beta-lactamases CDD-1 and CDD-2 from *Clostridium difficile*. We have rewritten this sentence as follows to avoid any misunderstanding:

DBL-homologs with a Tat-SP seem to be more specific to Alphaproteobacteria (*Bradyrhizobium*) whereas the "OTHER-SP" prediction is mainly associated with intrinsic pseudo-DBLs (CDD-1 and CDD-2 enzymes) of *Clostridioides* (29).

Line 504: A verb is missing. Perhaps "..., of which 34% were DBL-homolog sequences."?

>>> This sentence was indeed unclear and has been rewritten as follows:

Around 28% of gene sequences belong to classified strains, of which 55% clinical strains and, among those "clinical genes", 73% are DBL-homolog sequences (Table 2).

Line 539: Replace "cluster" with "clusters"

>>> Done. Thank you.

Line 547: Consider replacing "unknown" with "unclassified" for consistency with Table S2.

>>> Thank you for highlighting this inconsistency. We changed it to "unclassified".

Lines 549-562: Perhaps state the conditions here again (although they are in the Materials & Methods): pET24a(+)-based plasmid with T7 promoter with induction (0.5 mM IPTG). This is important, because it represents an unnatural overproduction condition, which could be responsible for proteins being expressed in the insoluble form (inclusion bodies?). Were any attempts made to refold those proteins? The residual activity of all enzymes listed in Table 3 against nitrocefin (the best substrates of the four tested) can probably be explained with a small portion of the proteins going into solution.

>>> As you requested, the following sentences were added to the paragraph:
The OXAVL01-10 genes were cloned in the pET24a(+) plasmid under the control of the strong T7 promoter and introduced in *E. coli*. The production of OXAVL01-10 was induced by IPTG and evaluated by SDS-PAGE and beta-lactam hydrolysis.

>>> We agree with your explanation. Indeed, OXAVL04, OXAVL08 and OXAVL10 were found in inclusion bodies. However, we did not attempt to refold them. The related sentence has been modified as follows:

For OXAVL04, OXAVL08 and OXAVL10, a large production of the beta-lactamases was found only in the insoluble fractions at both culture temperatures, likely indicating the formation of inclusion bodies.

>>> A sentence "These results may only be indicative of the true spectrum of activity because of the low fraction of soluble enzymes present in some cases" has been added to better contextualize the crude extract experiments.

Line 731: change "represent" to "represents"

>>> Done. Thank you.

Line 922: replace "broken" with "broken down"

>>> Done. Thank you.

Reviewer 2

Data are already publicly available. A link is provided in the Data Availability section of paper (lines 327 - 329). <https://doi.org/10.6084/m9.figshare.18544955>

I found this study and paper to be technically sound, with enough detail to reproduce the study. It is an interesting exploration of Class D beta-lactamases and their hosts. In the attached review are grammatical suggestions for clarification.

>>> Thank you for your positive comments regarding our study. We have addressed all your grammatical suggestions from the attached manuscript. However, we did not understand your suggestions for lines 219 and 339 and, therefore, we have not changed anything for these lines yet.

February 20, 2022

Prof. Denis Baurain
University of Liège
InBioS-PhytoSYSTEMS, Eukaryotic Phylogenomics
Quartier Vallée 1
chemin de la Vallée 4
Liège, Liège B-4000
Belgium

Re: Spectrum00315-22R1 (An extended reservoir of class-D beta-lactamases in non-clinical bacterial strains)

Dear Prof. Denis Baurain:

Your manuscript has been accepted, and I am forwarding it to the ASM Journals Department for publication. You will be notified when your proofs are ready to be viewed.

Sincerely,

Monica Garcia-Solache
Editor, Microbiology Spectrum